# Epidemiology of Echovirus 30 Infections Detected in a University Hospital in Catalonia, Spain, in 1995–2020

**DOI:** 10.3390/microorganisms10030592

**Published:** 2022-03-09

**Authors:** Margarita del Cuerpo, Jon Gonzalez de Audicana, Maria Dolores Fernandez-Garcia, Pilar Marín, Montserrat Esteban, Montserrat Español, María Cabrerizo, Núria Rabella

**Affiliations:** 1Microbiology Department, Hospital de la Santa Creu i Sant Pau, Sant Pau Biomedical Research Institute (IIB Sant Pau), 08041 Barcelona, Spain; pmarin@santpau.cat (P.M.); mesteban@santpau.cat (M.E.); mespanol@santpau.cat (M.E.); nrabella@santpau.cat (N.R.); 2Departament de Genètica i Microbiologia, Universitat Autònoma de Barcelona (UAB), 08193 Bellaterra, Spain; 3Enterovirus and Viral Gastroenteritis Unit, National Centre for Microbiology, Instituto de Salud Carlos III, 28222 Madrid, Spain; jonma_audikana@hotmail.com (J.G.d.A.); mdfernandez@isciii.es (M.D.F.-G.); mcabrerizo@isciii.es (M.C.); 4Centro de Investigación Biomédica en Red de Epidemiología y Salud Pública (CIBERESP), Instituto de Salud Carlos III, 28029 Madrid, Spain

**Keywords:** enteroviruses, echovirus 30, central nervous system, cerebrospinal fluid, phylogenetic analysis, epidemiology

## Abstract

There is a growing interest in echovirus 30 (E30), an enterovirus responsible for neurological disease and hospitalization. There are multiple studies of outbreaks, but few that study the epidemiology over long periods of time. Our study aims to describe the clinical, epidemiological and microbiological characteristics of a series of E30 infections detected over 26 years. Data were retrospectively collected from a database of all enterovirus infections identified in our laboratory. They were detected by viral isolation or nucleic acid detection in patients presenting with respiratory or neurological infections, rash, sepsis-like syndrome, or gastroenteritis. Enterovirus genotyping was performed by amplification of the VP1 gene using RT-nested PCR, followed by sequencing and BLAST analysis. Of the 2402 enterovirus infections detected, 1619 were linked to at least one genotype and 173 were caused by E30. Clinical information was available for 158 (91.3%) patients. E30 was associated with neurological infection in 107 (67.8%) cases and it was detected almost every year. Phylogenetic analysis was performed with 67 sequences. We observed that E30 strains circulating in Catalonia from 1996 to 2016 belong to two lineages (E and F), although the majority cluster was in F. In 2018, lineage I emerged as the dominant lineage.

## 1. Introduction

Enteroviruses (EVs) belong to the *Picornaviridae* family, which currently consists of 68 genera and 158 species. The genus *Enterovirus* consists of 15 species, of which 4 EV species (A, B, C, D), with more than 100 different genotypes, and 3 rhinovirus species (A, B, C) can infect humans. They are small (24–30 nm) non-enveloped viruses, consisting of a single strand of positive polarity RNA acting directly as messenger RNA, and an icosahedral capsid consisting of four proteins (VP1, VP2, VP3, VP4) [1]. Genetic differences in the VP1 polyprotein give rise to the different genotypes [2,3]. Mutations and recombination events are frequent in EVs. Exchanges can occur within the same genotype or between different genotypes, contributing to the creation of variants and the evolution of EVs [4,5].

EVs are among the most common viruses infecting humans and are distributed worldwide. Some genotypes are endemic with minor variations in circulation from year to year and others are epidemic in nature, causing outbreaks in certain years. They are transmitted predominantly by the fecal–oral route but also by respiratory aerosols and contact with vesicular fluid and surfaces contaminated with the virus.

Up to 90% of EV infections are asymptomatic, but they can cause a wide spectrum of diseases, ranging from a mild non-specific febrile syndrome, colds, and muco-cutaneous rashes, such as herpangina and hand-foot-mouth disease (HFMD), to severe illnesses, such as pneumonia, myocarditis, pericarditis, aseptic meningitis, encephalitis or acute flaccid paralysis (AFP), which may even lead to death. They are also involved in sepsis and congenital and neonatal infections. Worldwide, EV infections account for hundreds of thousands of hospital admissions, with aseptic meningitis being the most common, and although they are a major cause of morbidity and mortality, especially in young children, notification of these cases is not compulsory in many countries, and therefore EV infection is likely to be underestimated [6,7,8].

No disease, clinical presentation or symptom can be attributed to a single genotype. However, different genotypes have been associated with certain clinical pictures, such as polioviruses (PVs) and poliomyelitis [9], enterovirus D68 (EV-D68) and severe respiratory and neurological presentations [10], enterovirus A71 (EV-A71) and HFMD and encephalitis [11], or echovirus 30 (E30), long described as a cause of aseptic meningitis worldwide [12,13,14,15,16].

E30 is an EV genotype frequently detected in our environment [17]; it belongs to the EV B species, is antigenically heterogeneous, and is classified phylogenetically into nine lineages [18,19]. E30 outbreaks show a cyclical incidence pattern of 3 to 5 years that is generally associated with the rapid spread of different strains [20,21,22].

Many studies describe outbreaks of aseptic meningitis caused by E30, but there are few about the epidemiology and molecular evolution of this genotype over a long period. Given the growing interest in E30 as a cause of neurological disease and hospitalization [23], our study aims to describe the clinical, epidemiological and microbiological characteristics of a series of E30 infections detected in our laboratory over a period of 26 years.

## 2. Materials and Methods

### 2.1. Study Design

Epidemiological, microbiological, and clinical data were retrospectively collected from all E30 infections identified in the virology laboratory of a university hospital (Hospital Santa Creu i Sant Pau, Barcelona) from 1995 to 2020. This hospital is a 548-bed reference medical center for highly complex pathologies that provide healthcare services for a population of 407,550. The laboratory is accredited as a sub-national polio laboratory by the World Health Organization (WHO) and has participated in a Spanish EV molecular surveillance network since 1998.

### 2.2. Clinical Samples

For routine diagnosis of EV/PV infections, the laboratory received annually a mean of 2180 clinical samples from patients, mainly pediatric, with the following diseases: respiratory, neurological, muco-cutaneous, sepsis-like or gastroenteric. The main types of biological samples were respiratory (nasopharyngeal aspirates, bronchoalveolar lavages, pharyngeal exudates), stools, cerebrospinal fluids (CSF) and skin exudates.

### 2.3. Methods

#### 2.3.1. EV Isolation

Until 2007, the only EV detection method used by the laboratory was the culture of clinical samples on different cell lines: human fetal lung MRC-5 (RD BIOTECH (Besançon, France), human lung carcinoma A549 (Vircell S.L. (Granada, Spain), human epidermoid laryngeal carcinoma HEp2 (Vircell S.L. (Granada, Spain) and human embryonal rhabdomyosarcoma RD (Enterovirus and Viral Gastroenteritis Unit. National Centre for Microbiology. Instituto de Salud Carlos III, Madrid, Spain). Samples were processed and inoculated following the WHO polio laboratory manual for PV/EV isolation [24,25]. Viral cultures were incubated at 37 °C and observed daily under inverted microscopy for 14 days to detect a cytopathic effect compatible with EV infection. An indirect immunofluorescence technique with a mixture of EV-specific monoclonal antibodies (Enterovirus Screening Set LIGHT DIAGNOSTICS™ and D^3^ IFA Enterovirus Identification Kit Diagnostic HYBRIDS) was used for confirmation of the EV infection.

#### 2.3.2. EV Detection

A commercial reverse transcription-polymerase chain reaction (RT-PCR) assay (XpertEV, Cepheid, Sunnyvale, CA, USA), designed to detect EV RNA directly in clinical samples in the 5′ UTR untranslated region of the genome, was adopted in 2007.

#### 2.3.3. EV Genotyping

EV-positive samples and isolates were sent to the Enterovirus Unit of the National Centre for Microbiology (Instituto de Salud Carlos III, Madrid, Spain) for EV type characterization. EV genotyping was performed by amplification of the 3′-part of the VP1 gene using conventional RT-nested PCRs specific for EV-A, B, C or D, as previously reported [26,27], followed by sequencing and BLAST analysis of the sequences obtained (http://blast.ncbi.nlm.nih.gov/). Specific type was assigned when the homology in the nucleotide sequence was higher than 75%.

#### 2.3.4. Phylogenetic Analysis

A phylogenetic analysis was carried out with 67 E30 available sequences detected between 1996 and 2018, the prototype strains (Bastianni, Frater) and 50 sequences in the same 3′-VP1 region (420 bps) from other countries available in GenBank. Alignments were performed with ClustalW [28] and the phylogenetic tree was built with the neighbor-joining method, the maximum composite likelihood distance model, and 1000 pseudo-repeats, such as the bootstrap method, in the MEGA X program [29]. All VP1 sequences were deposited in GenBank. Genome sequences obtained in this study were deposited in GenBank (accession numbers OM925759-OM925825).

#### 2.3.5. Ethics Statement

The clinical and epidemiological data were treated anonymously and their study was approved by the clinical research ethics committee of the Institut de Recerca de l’Hospital de la Santa Creu i Sant Pau—IIB Sant Pau with protocol code IIBSP-ENT-2019-42.

## 3. Results

### 3.1. EV Detection and Genotyping

During the 26-year study period, 2402 patients were diagnosed with EV infection. A total of 2929 EV-positive samples were included in this study, and 2033 (69.4%) of these were genotyped. The specific EV type was determined in 1619 patients. Four predominant genotypes were found: echovirus 11 (E11) in 199 cases (12.3%), echovirus 6 (E6) in 186 cases (11.5%), coxsackievirus B5 (CVB5) in 124 cases (7.6%) and E30 in 173 cases, representing 10.7% of the total genotyped EV.

### 3.2. Demographic, Epidemiological and Clinical Characteristics of E30 Infections

The male/female ratio of the E30-positive patients was 1.5:1 (106/67) with a mean age of 8 years, ranging from 7 days to 81 years. Of the total E30 infections, 87 (50.4%) were in children between 0 and 3 years, with infants up to 3 months of age accounting for 16% of the total patients. Clinical information from 158 (91.3%) patients was available. E30 infections associated with the central nervous system (CNS) were 107/158 (67.8%), with most of them (103, 96.3%) with symptoms suggestive of aseptic meningitis, while only 4 cases were diagnosed with meningoencephalitis or encephalitis. Non-specific febrile syndrome (or without further information) was found in 34/158 (21.5%) cases, respiratory diseases in 12/158 (7.6%) and muco-cutaneous lesions in 5/158 (3.1%) cases (Table 1).

A total of 254 samples from these 173 patients were studied. Two hundred and thirty-two samples were E30 positive, of which 87 (37.5%) were CSF, 99 (42.7%) were respiratory (mainly nasopharyngeal aspirate mucus and/or pharyngeal exudates and one bronchoalveolar lavage), 42 (18.1%) were fecal, 2 were urine and there were 2 skin exudates (0.9% each) (Table 2).

In 48 patients (27.8%) multiple samples were studied, with CSF, respiratory and stool samples being the most frequent combination (22 patients) (Table 2). In the remaining 125 patients (72.2%) only one type of sample was studied. In 38 patients with CNS involvement, more than one type of specimen was studied. In these cases, the detection of E30 in CSF was efficient in 76% (29/38) of the cases, in respiratory specimens in 80% (27/34) and in feces in 100% (26/26).

In addition to E30 infection, other viruses were detected in 10 (5.8%) patients, in the same or a different sample. In 6 cases, other EV types were identified: in one patient E6 was co-detected in CSF; in another, E30 was found in CSF and feces and E11 in the respiratory sample; in two cases, E30 and EV-A71 were co-detected in feces; one E30-positive patient in feces had an E11 in the respiratory sample; finally, in one patient, E30 was detected in urine and E11 in the respiratory sample. In addition, other respiratory viruses (two adenoviruses-ADV-, one parainfluenza 3 virus -PIV3-, and one respiratory syncytial virus -RSV-) were found in four cases.

E30 was detected in almost all the years recorded, with an increase in incidence every 2–5 years, which did not always coincide with the years of highest observation of overall EV infections. The year with the highest number of E30 detections in this study was 2007, with other upsurges in 1996, 2000-01, 2005, 2012-13, 2015-16 and 2018. Outbreaks sometimes occurred in a single year or were extended over a 2-year period (Figure 1, Table 3). Even in the years with the highest detection of E30, this did not usually exceed 10–35% of all the EV infections detected, except in 2007 when E30 cases made up more than 50% of the total EV detected.

Regarding the seasonal distribution of E30 infection, although the virus was detected in practically all months of the year, there were two periods of increased detection: one between May and July, which accounted for almost half of all cases, and another shorter period in autumn (November and December). Comparing the number of cases associated with CNS infection with the rest of the EV infections, we find that in the months of June and July, the number of patients with neurological involvement is three times higher than in the rest of the year, while in the autumn peak (November, December), that number increases in parallel with the other infections, and in the winter months (January, February), it is surpassed by the other infections (Figure 2).

### 3.3. E30 Phylogenetic Analysis

To investigate the spatiotemporal relationships among E30 strains detected in our hospital from 1995 to 2020, 67 available VP1 sequences were compared to other homologous strains from different countries worldwide. According to the phylogenetic classification in 9 lineages (A–I), previously published [18,20] sequences from 1996 to 2006 belonged to E and F, and most of the E30 detected between 2007 and 2016 were grouped in F, in which different subclades can be distinguished. Finally, all VP1 sequences from 2018 clustered within lineage I, with a higher nucleotide divergence from C–H (26.2–35.5%). No sequences from Catalonia belonged to A–D and G–H lineages (Figure 3).

## 4. Discussion

This study presents the epidemiology and molecular evolution of the E30 detected in a university hospital in Barcelona (Catalonia, Spain) over a period of 26 years.

E30 has been widely detected for decades and is described as one of the most predominant EV genotypes in multiple reports [20,30,31,32]. In our study, it is the third most detected EV type, after E6 and E11, and was found in approximately 11% of cases. However, our percentage is slightly lower compared to data for the entire Spanish territory [17,33], where E30 is the most prevalent genotype and is reported to exceed 25%.

We found that in almost 68% of cases with E30 detection, neurological involvement was recorded, almost entirely meningitis. This result was to be expected, as E30 is one of the genotypes most frequently associated with aseptic meningitis [34,35], with multiple outbreaks widely documented throughout the world. However, we also detected E30 related to other clinical manifestations, such as non-specific fever and respiratory symptoms. Milder illnesses, such as uncomplicated respiratory infections, may be underestimated because clinical samples from these pathologies are not routinely collected for testing.

As is the case for infections with other EV genotypes [36], E30 was detected mainly in children: 50% of the patients were under three years old, with a high incidence (16% of the total) in babies from 0 to 3 months. This could be because babies and very young children are more prone to infection as they have not yet acquired immunity, and medical attention is sought quickly for symptoms, such as fever or irritability in this population. Some studies with patients of all ages report that in addition to the pediatric age group, young adults also have a high incidence of infection [20,37] but this was not observed in our series.

We found that in our region, E30 can be considered an endemic EV since it circulates almost every year (without detection only in 6 out of 26 years), with an increase in cases every 2–5 years in which the number of E30 infections detected ranges from 15% to 54% of the total EVs, with an outbreak duration of 1 to 2 years, in accordance with other studies. It is possible that after periods of intense activity, in which population immunity is created for circulating variants, detections drop until different variants break out, causing a significant increase in cases to reappear [20,21,22]. In our study, the emergence of the new F and I lineages correlates with an increase in the incidence in Catalonia in the years 2005–2007 and 2018, respectively.

EV infections present seasonality in temperate zones, being more frequent in summer and autumn, and a homogeneous distribution throughout the year in tropical and subtropical zones [38]. In our series, E30 detections follow a seasonal pattern, with a considerable increase in cases in late spring and early summer, but we also found a small increase in autumn (November and December). If we separate the cases with neurological symptoms, we observe a large peak in June and July, in accordance with previous reports of aseptic meningitis outbreaks [13]. However, if we look only at non-neurological cases, the seasonality is different, with three peaks in February, May, and November-December. It is possible that this increase is due to the fact that in the autumn and winter months, respiratory infection is more frequent than neurological infection. However, more studies with a greater number of cases of respiratory infection due to E30 would be needed to confirm this hypothesis.

In almost 6% of E30 cases, other viruses were co-detected, which was favored by the study of multiple samples per case. The most frequent association was between E30 and other EV types (E6, E11 and EV-A71), and respiratory samples were found to have the greatest viral diversity. The versatility of the technique used to detect EV in respiratory samples, cell culture in various cell lines, allowed other conventional respiratory viruses (ADV, RSV, PIV3) to be detected in 4 cases. It should be noted, that in one case we found two EVs (E30 and E6) in the CSF specimen of an elderly man. Coinfections with more than one virus are widely described in the literature [39,40,41,42]. Some authors suggest that simultaneous infection by more than one virus generates more severe symptoms [43] and that infection by more than one genotype can generate recombination events between different species [4,44], giving rise to new variants. In our series, the number of co-detections is too small to draw conclusions, and it would be of interest to conduct more studies with a larger number of cases to investigate the clinical importance of coinfection.

Given that the finding of the virus in CSF samples confirms the neurological infection, it is the most studied sample type in the numerous studies on outbreaks of meningitis caused by E30. We found that CSF is a good sample to detect E30 in aseptic meningitis (90% of the CSF samples studied were positive) compared to other genotypes found in cases of neurological involvement, such as EV-A71 [11,45,46] or EV-D68 [27] where they are barely detected. However, these last two genotypes are more associated with encephalitis or AFP than with meningitis [35], which could be the reason why the virus was not found in the CSF at the time of the study [47]. It should be noted that in our series, 10% of the CSF samples were negative, and the infection was diagnosed by the analysis of other clinical samples, such as nasopharyngeal mucus and/or feces, where the efficiency of nasopharyngeal exudate or feces was 80% and 100%, respectively. It is of interest to study samples other than CSF in cases where neurological symptoms are present.

The resulting phylogenetic tree showed a clear turnover of E30 lineages over the sampled years, in accordance with other studies [18,19,20]. A recent study analyzing the evolution of E30 strains in Spain observed a genetic turnover in which an emerging lineage I replaced previous lineages circulating in Spain. Moreover, this study observed that lineage F contained most of the E30 strains previously identified in Spain from 1996 to 2018 [19]. In agreement with these observations, the majority of strains in the present study (from 1997 to 2016) clustered in lineage F. Several sub-lineages could be distinguished in this lineage, which correlated with an increase in E30 incidence in 2007, 2013 and 2015-16. We similarly note that all sequences from 2018 belong to the emerging lineage I. According to Benschop et al. [20], the E30 upsurge observed in several European countries during 2018 was caused by the appearance of two different clades, G1 and G6 (corresponding to lineages E and I, respectively). In this study, two analyzed sequences obtained from Catalonia in 2018 (MK815445 and MZ389230) clustered in these two respective clades. It is noteworthy, that in our study all sequences from 2018 belonged only to lineage I, a finding in accordance with Gambaro et al. [19], who reported that E30 strains from 2018, sampled in Andalusia (South Spain), all belonged to lineage I. Although these phylogenetic analyses show the emergence of lineage I in Spain, overall, E30 was not the most frequent EV genotype detected that year, being third in predominance after EV-D68 and E9, respectively, and in 2019, it accounted for 14.5% of total typed EV (data not published).

It is possible that the present study suffered from a selection bias as the majority of individuals diagnosed with EV infection were patients who attended our hospital with clinical symptoms significant enough to warrant a hospital consultation and for physicians to collect clinical samples for laboratory analysis.

## 5. Conclusions

In the region of Catalonia, E30 is an endemic EV that circulates almost every year with a peak of cases in late spring and early summer. E30 was detected from CSF specimens of most of the patients with neurological involvement, but not in all. The phylogeny showed that E30 strains circulating in Catalonia shifted from lineages E and F to lineage I in 2018. The emergence and dominance of a single lineage in 2018, replacing previous dominant lineages, highlights the importance of continuous surveillance of E30 circulation as well as other clinically relevant EVs.

## Figures and Tables

**Figure 1 microorganisms-10-00592-f001:**
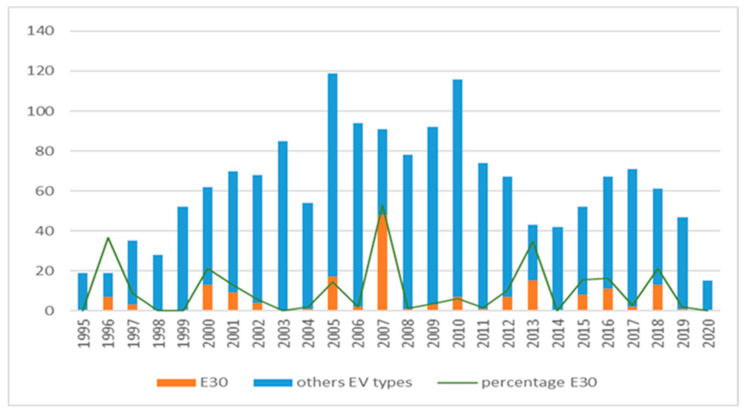
Annual distribution of E30 detections vs. other EV genotypes.

**Figure 2 microorganisms-10-00592-f002:**
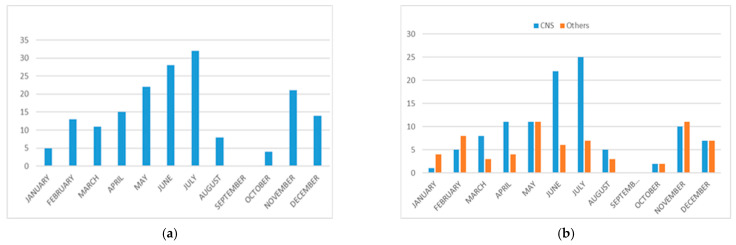
(**a**) Seasonal distribution of all E30 infections included in this study; (**b**) seasonal distribution of E30-positive cases with central nervous system (CNS) diseases versus other clinical presentations.

**Figure 3 microorganisms-10-00592-f003:**
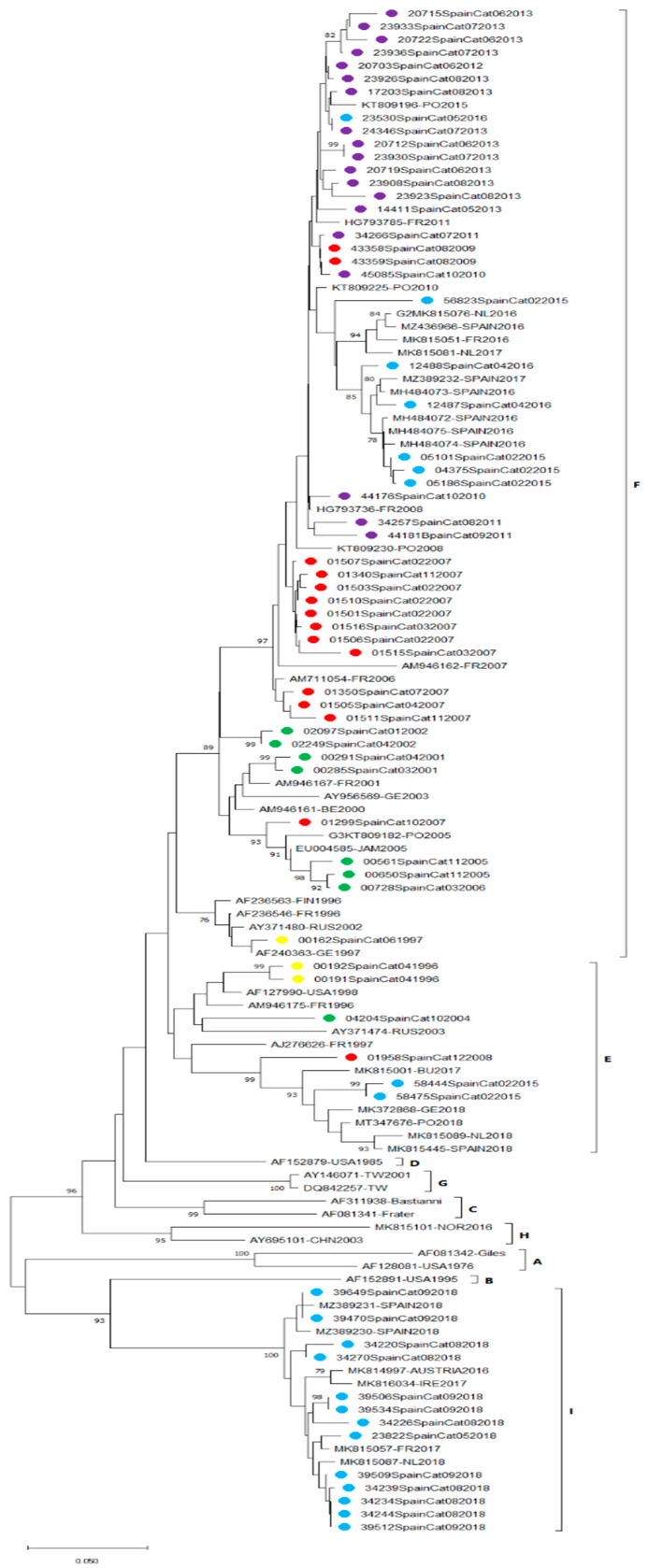
Phylogenetic tree of the 3′-VP1 region with E30 sequences from 1996–2020 classified into 9 lineages, A–I. The tree was reconstructed using the neighbor-joining method and the maximum composite likelihood model. The tree was rooted with the prototype sequences Bastianni (AF311938) and Frater (AF081341). Bootstrap resampling (1000 replicates) was used to determine grouping robustness. Values of >70% are shown. E30 sequences obtained in this study are colored according to the year of detection (yellow 1996–1997; green 2001–2006; red 2007–2009; violet 2010–2013; blue 2015–2018).

**Table 1 microorganisms-10-00592-t001:** Demographic and clinical characteristics of 173 patients with E30 infection.

Age Group	E30 Infections	Percentage (%)
0–3 months	28	16.2
4–23 months	25	14.5
2–3 years	34	19.7
4–5 years	24	13.9
6–10 years	28	16.2
11–17 years	10	5.8
18–40 years	16	9.2
>40 years	6	3.4
not specified	2	1.1
**Sex**		
Female	67	38.7
Male	106	61.3
** Patients with clinical information **	** 158 **	** 91.3 **
Non-specific febrile syndrome	34	21.5
Exanthema	5	3.1
Respiratory symptoms	12	7.6
Neurological symptoms	107	67.8
Meningitis	103	96.2
Meningoencephalitis	2	1.9
Encephalitis	2	1.9

**Table 2 microorganisms-10-00592-t002:** Types of samples studied in 173 cases of E30 infections.

Sample		*n*		E30-Positive	Percentage (%)
CSF *		97		87	89.7
Respiratory sample		108		99	91.7
Feces		42		42	100
Urine		5		2	40
Skin exudates		2		2	100
Total		254		232	91.3
**Cases with multiple specimens analyzed (*n* = 48)**					
			**Cases**	**Percentage (%)**	
CSF + RS ** + feces + urine			1	0.6	
CSF + RS + urine			2	1.1	
CSF + RS + feces			22	12.7	
CSF + RS			10	5.8	
CSF + feces			3	1.7	
RS + feces			8	4.6	
RS + feces + urine			1	0.6	
RS + urine			1	0.6	

* CSF, cerebrospinal fluid; ** RS, respiratory sample.

**Table 3 microorganisms-10-00592-t003:** E30 infections with respect to the total of EV genotyped by year.

Year	EV Detection	Genotyped EV (%)	E30 Detection (%)
1995	23	19 (82.6)	0
1996	42	19 (45.2)	7 (36.8)
1997	76	35 (46)	3 (8.6)
1998	76	28 (36.8)	0
1999	115	52 (45.2)	0
2000	111	62 (55.9)	13 (21)
2001	93	70 (75.3)	9 (12.9)
2002	98	68 (69.4)	4 (5.9)
2003	96	85 (88.5)	0
2004	74	54 (73)	1 (1.8)
2005	151	119 (78.8)	17 (14.3)
2006	132	94 (71.2)	2 (2.1)
2007	137	91 (66.4)	48 (52.7)
2008	129	78 (60.5)	1 (1.3)
2009	119	92 (77.3)	3 (3.3)
2010	162	116 (71.6)	7 (6)
2011	122	74 (60.6)	1 (1.4)
2012	130	67 (51.5)	7 (10.4)
2013	63	43 (68.2)	15 (34.9)
2014	64	42 (65.6)	0
2015	70	52 (74.3)	8 (15.4)
2016	91	67 (73.6)	11 (16.4)
2017	92	71 (77.2)	2 (2.8)
2018	72	61 (84.7)	13 (21.3)
2019	48	47 (98)	1 (2.1)
2020	16	15 (93.7)	0
Total	2402	1619 (67.4)	173 (10.7)

## Data Availability

Not applicable.

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
