# Peer review of "Epidemiology of Echovirus 30 Infections Detected in a University Hospital in Catalonia, Spain, in 1995–2020"

_microorganisms, 2022, doi:10.3390/microorganisms10030592_

Round 1

Reviewer 1 Report

This review of enterovirus B30 (EV-B30) isolations identified in a 26 year period by the virology laboratory of a university hospital in Barcelona, Spain is a thorough and well written article.  In examination of 2402 enterovirus infections, 173 were caused by EV-B30 which gives a somewhat limited group of detections to analyze.  Nevertheless, there were a few findings that were interesting.  One was the confirmation that the lineages of the isolates defined by Bailley et al and Benschop et al, show the lineage F 1997-2016 and then Lineage I appearing in 2018.  Another is that there is a high correlation of EV-B30 in these cases with neurological disease, primarily meningitis and that detection in CSF was efficient in those cases.  There was also an interesting finding of an additional seasonal peak in EV-B30 beyond the normal spring to early summer, in November and December.  Of course, the fact that there is a geographical limitation to the data can give undue strength to the seasonal findings. 

I think this work confirms many EV-B30 and enterovirus findings.  It is not particularly novel but a worthwhile epidemiologic study.  As the manuscript is clearly written and well researched, I think it should be published. 

Author Response

The text has been reviewed by a native professional proof-reader.

Reviewer 2 Report

The authors described on the clinical, epidemiological, and microbiological characteristics of a series of E30 infections detected at a University Hospital in Catalonia, Spain between over a period of 26 years.   As they describe, there are few studies about the epidemiology of E30 over long period and this manuscript shows us its characteristics; E30 is the third most detected EV type behind E6 and E11, neurological involvements were recorded in almost 68% of cases with E30 detection, and E30 is an endemic EV that circulates almost every year with an increase in cases every 2-5 years.   I respect their huge data.   My comments are as follows.

1.(Introduction 6th paragraph Page2) E30 is an EV genotype frequently detected…and is antigenically heterogeneous.   Were there any differences of antigenicity among genotypes E, F and G E30 strains in Spain?  You also mentioned about immunity in the 5th paragraph in Discussion.

2.(Materials and methods 2.3.4 Phylogenetic analysis Page3) How did you choose the 67 E30 strains for the phylogenetic analysis?

  1. (Materials and methods 2.3.4 Phylogenetic analysis Page3) “3’-VP1 region” means 420 nucleotides long as described in the reference 26 (Cabrerizo M et al)? Phylogenetic tree was constructed based on these 420 bps?

  1. (Materials and methods 2.3.4 Phylogenetic analysis Page3) The authors described that all VP1 sequences were deposited in GenBank. Where are their GenBank number? It is likely that there is no GenBank number in the Figure 3.

  1. (Table 1) There are 6 patients > 40 years old. I am interested in their clinical symptoms. Did they have neurological symptoms?  In the 7th paragraph in discussion, you described that you detected E30 in the CSF of an elderly man.

  1. (Figure 1 and Table 3) It is likely that information in Figure 1 and Table 3 is duplicated.

  1. (Figure 2) Figure 2 (a) is necessary?

8.(Discussion 2nd paragraph from the bottom Page 10) Accession number MZ389230 is not shown in the phylogenetic tree in Reference 19.   It is shown as “LCR519”.  Thus, it was very difficult for me to find the strain.

9.(Discussion 2nd paragraph from the bottom Page 10) G1 and G6 (Reference 20) are corresponding to lineages E and I in this study.   I hope representative strains of G1 and G6 lineages in Reference 20 are included in Figure 3.   I also hope that information such as “G1 and G6 (Reference 20) are corresponding to lineages E and I in this study” is shown in Figure 3 or Figure Legend.

  1. (Conclusions) The authors described that E30 is an “endemic” EV. I means that lineage I strains evolved from lineage E and/or F strains in Spain?

Author Response

The answer is in the file 
